# MATTEXT: LANGUAGE MODELS NEED MORE THAN TEXT & SCALE FOR MATERIALS MODELING

## ABSTRACT

Effectively representing materials as text has the potential to leverage the vast advancements of large language models (LLMs) for discovering new materials. While LLMs have shown remarkable success in various domains, their application to materials science remains underexplored. A fundamental challenge is the lack of understanding of how to best utilize text-based representations for materials modeling. This challenge is further compounded by the absence of a comprehensive benchmark to rigorously evaluate the capabilities and limitations of these textual representations in capturing the complexity of material systems. To address this gap, we propose MatText, a suite of benchmarking tools and datasets designed to systematically evaluate the performance of language models in modeling materials. MatText encompasses nine distinct text-based representations for material systems, including several novel representations. Each representation incorporates unique inductive biases that capture relevant information and integrate prior physical knowledge about materials. Additionally, MatText provides essential tools for training and benchmarking the performance of language models in the context of materials science. These tools include standardized dataset splits for each representation across a range of dataset sizes, probes for evaluating sensitivity to geometric factors, and tools for seamlessly converting crystal structures into text. Using MatText, we conduct an extensive analysis of the capabilities of language models in modeling materials with different representations and dataset scales. Our findings reveal that current language models consistently struggle to capture the geometric information crucial for materials modeling across all representations. Instead, these models tend to leverage local information, which is emphasized in some of our novel representations. Our analysis underscores MatText's ability to reveal shortcomings of text-based methods for materials design.

## 1 INTRODUCTION

Large language models (LLMs) have been increasingly applied to scientific disciplines (Miret & Krishnan, 2024; Mirza et al., 2024; AI4Science & Quantum, 2023; Madani et al., 2023). This includes applying LLMs to materials modeling (Xu et al., 2023; Rubungo et al., 2023; Jablonka et al., 2023; 2024) such as diverse property prediction tasks and the generation of new materials based on text-based representations (Gruver et al., 2023; Flam-Shepherd & Aspuru-Guzik, 2023; Antunes et al., 2024). Despite recent efforts, there is no clear understanding of the performance and limitations of language models for text-based material-property prediction. For language modeling, it is commonly believed that increasing the number of parameters or data improves performance in various tasks. In the field of material science, however, physical laws govern the relationship between materials and their properties. As such, the scaling laws may differ from those for the modeling of natural language.

That lack of understanding related to scaling laws is currently limiting progress. While there are practically an infinite number of ways to represent materials in text, the research community currently has little understanding of how to design an effective representation. This shortcoming is further aggravated by the absence of tooling to reliably create text-based representations of materials, as well as datasets and benchmarks to evaluate language model performance on materials modeling tasks. Moreover, existing datasets related to text-based materials modeling are fragmented and

heterogeneous in their representation and availability, and can hence not be used for systematic benchmarking.

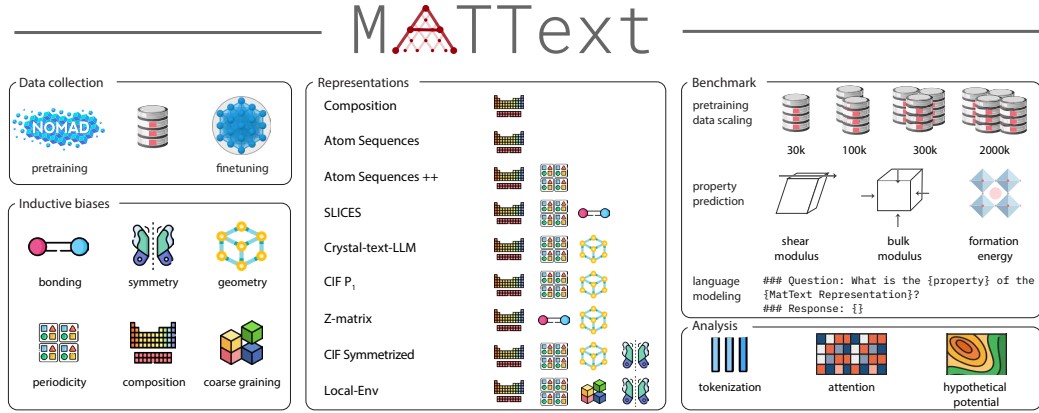

Figure 1: Overview of the MatText framework. For MatText, we compile pretraining and fine-tuning data and implement text representations of crystal structures with different inductive biases in a standardized, object-oriented framework. We use the data and text representations to build a property-prediction benchmark for language modeling of materials properties and analyze the models and results to reveal the limitations of current approaches.

In this work, we propose a comprehensive dataset and benchmark that spans nine text-based representations of solid-state materials, including five novel representations first introduced in MatText. Concretely, we make the following contributions (Figure 1).

- **MatText Benchmark:** We report the most comprehensive evaluation of text-based material property prediction. The benchmark unifies four previously reported representations and five new representations we propose in this work, covering many relevant inductive biases. The MatText benchmark is based on open-source datasets and established materials informatics tooling (Dunn et al., 2020; Ong et al., 2013) and can be easily applied to test other systems.
- **MatText Representations:** To enable the MatText benchmark, we also develop a software package to convert geometric representations of materials into text-based representations. Besides transforming crystal structures into text, our framework also facilitates the translation into tokens by providing reference implementations of tokenizers, as well as decoding and robustness evaluation utilities.
- **MatText Analysis:** We analyze the current shortcomings of language models related to materials property modeling. Our analysis spans representations, as well as multiple data and model scales and architectures, and highlights the importance of locality as inductive biases. We provide systematic evidence that current language modeling frameworks falter to effectively leverage geometric information about crystal structures.

Our observations suggest that scaling current language models similar to LLMs for natural language may not improve material property predictions. We believe that our MatText framework will enable the design and evaluation of better modeling frameworks.

## 2 RELATED WORK

**Language Models as Supervised Learners**    (Large) language models (Brown et al., 2020; Touvron et al., 2023; Rae et al., 2021; Hoffmann et al., 2022; Raffel et al., 2023) have gained substantial attention due to their exceptional performance on diverse tasks (Bubeck et al., 2023; Wei et al., 2022), including supervised learning problems. They are typically trained by self-supervised pretraining on a large corpus before supervised fine-tuning on labeled data (Howard & Ruder, 2018). Encoder models such as BERT (Devlin et al., 2018) have been augmented with classification or regression heads to perform classification or regression tasks based on an input sequence (Adhikari et al., 2019).

Dinh et al. (2022) showed that LLMs can be fine-tuned without architecture changes (such as adding a regression head) to solve classification and regression tasks based on tabular data inserted into text templates. Vacareanu et al. (2024) demonstrated that pre-trained large language models can perform such linear and non-linear regression using only in-context learning.

**Text-Based Modeling of Molecules, Proteins & Materials** Language modeling has become a popular approach for predicting protein structures and functions. The amino acid sequence is the foundation for the structure and function of a protein and can be easily represented as text (Ruffolo & Madani, 2024; Rives et al., 2021; Lin et al., 2023; Elnaggar et al., 2022; Xu et al., 2023). While recent research by Vig et al. (2021) has demonstrated that language models can capture structural information when trained on sequence data, other findings indicate that on many downstream tasks, the performance of protein language models does not scale with pretraining (Li et al., 2024).

For molecules, various text-based representations such as SMILES (Weininger, 1988) and SELFIES (Krenn et al., 2020; Cheng et al., 2023a) have been developed and used for language modeling (Krenn et al., 2022; Bran & Schwaller, 2023; Cadeddu et al., 2014; Frey et al., 2023; White, 2023; Noutahi et al., 2023). These representations have been successfully applied to tasks such as retro- and forward-synthesis (Pesciullesi et al., 2020; Schwaller et al., 2019), molecular property prediction (Chithrananda et al., 2020; Wang et al., 2019; Ahmad et al., 2022; Balaji et al., 2023), and conditional generation of molecules (Born & Manica, 2023; Bagal et al., 2021; Grisoni, 2023; Ghugare et al., 2023).

While successes in protein and molecular text representations provide inspiration, materials science poses unique challenges. Many properties, for example, depend on the 3D structure and periodic repetition of a unit cell motif (Hoffmann, 1987) making representing materials as text more challenging. Nevertheless, past work has proposed different representations that include distinct inductive biases, which we describe in greater detail in Section 3.1. For instance, Ganose & Jain (2019) aimed to create human-readable descriptions by proposing a tool to automatically generate natural-language descriptions of crystal structures, which have been used to create predictive embeddings and models (Qu et al., 2024; Sayeed et al., 2023; Rubungo et al., 2024; Korolev & Protsenko, 2023). For specific materials classes, such as metal-organic frameworks (MOFs), specialized representations like MOFid Bucior et al. (2019) have been developed and used for materials design Cao et al. (2023). However, unlike proteins, organic reactions, or small molecules, no natural representation has emerged for materials, making language modeling more challenging in this field.

**Inductive Biases for Material Modeling** The modeling of physical systems can often benefit from the inclusion of physical background knowledge as inductive bias. Locality, smoothness, and symmetry are the most widely used inductive biases (Musil et al., 2021). Locality is commonly incorporated using a distance cutoff and rationalized with the nearsightedness principle of quantum mechanics (Prodan & Kohn, 2005). Related to this is using coarse-grained molecular motifs as inductive bias (Sommer et al., 2023; Cheng et al., 2023b). Symmetry has been incorporated in many of the most performant models by designing invariant or equivariant features (Langer et al., 2022; Musil et al., 2021) or model architectures (Batatia et al., 2022; Satorras et al., 2021; Batzner et al., 2022; Thomas et al., 2018). Previous work has indicated that for certain phenomena (e.g., when all structures in a dataset are in the ground state), composition might implicitly encode geometric information (Tian et al., 2022; Jha et al., 2018; Wang et al., 2021).

## 3 BENCHMARK

Our dataset leverages open-source materials structures sourced from the NOMAD archive (Draxl & Scheffler, 2018; Miret et al., 2023). We interface with the developments from the `MatBench` benchmark suite (Dunn et al., 2020) for property prediction tasks and use our new MatText Python package to derive text representations.

Overall, we designed the MatText framework to allow the seamless conversion of crystals into text representations with many different inductive biases (Section 3.1). For this, we implemented several previously described representations and developed and implemented novel ones to cover additional inductive biases, such as coarse-graining, and to allow for the ablation of different design choices in text representations.

## 3.1 MatText Representations

The MatText subpackage focussed on representations is based on `pymatgen` (Ong et al., 2013) and provides an object-oriented way to convert crystal structures into text representations. Additionally, MatText provides canonical implementations of relevant tokenizers that can be directly used for diverse language model architectures. Moreover, MatText provides utilities to assess the robustness of representations with respect to permutations, perturbations, and translations. For invertible representations, MatText also implements decoders to convert text representations into `pymatgen` crystal structure objects. MatText is easy to extend and use and described with tutorials as well as code for all our analyses under MIT license. An example of usage is also shown in Appendix A.2.

Table 1: MatText Representations: The table summarizes the representations in this work, including existing and newly proposed representations, alongside the various inductive biases encoded in the text representations. Bonding captures the connection between atoms. Geometry describes the spatial arrangement of atoms. Symmetry describes that the representation incorporates information on the material's translational, rotational, and reflection symmetries. Periodicity refers to information about the periodic repeating unit. Coarse-graining refers to the aggregation of connected building blocks.

| | Stoichiometry | Bonding | Geometry | Symmetry | Periodicity | Coarse Graining | Reference |
|---|---|---|---|---|---|---|---|
| **Composition** | ✔ | | | | | | Tian et al. (2022) |
| **SLICES** | ✔ | ✔ | | | ✔ | | Xiao et al. (2023) |
| **CIF P$_1$** | ✔ | ✔ | ✔ | | ✔ | | Flam-Shepherd & Aspuru-Guzik (2023) |
| **Crystal-text-LLM** | ✔ | | ✔ | | ✔ | | Gruver et al. (2023) |
| **New Representations** | | | | | | | |
| **Atom Sequence** | ✔ | | | | | | - |
| **Atom Sequence++** | ✔ | | | | ✔ | | - |
| **CIF Symmetrized** | ✔ | | ✔ | ✔ | ✔ | | - |
| **Z-Matrix** | ✔ | ✔ | ✔ | | | | - |
| **Local-Env** | ✔ | ✔ | | ✔ | | ✔ | - |

In the subsequent section, we describe the representations used in our analysis along with the inductive biases. Examples for each representation are listed in Table 3.

The text representations have different information content (going from just composition information to information about the composition and the position of all the atoms), allowing us to analyze what information language models can use for material property predictions. In addition, the representations feature different combinations of inductive biases, which allows us to identify the most meaningful ones in our analysis.

### 3.1.1 Previously Reported Material Representations

**Composition** Prior work has shown that in certain cases, material composition alone can be predictive for various materials properties (Tian et al., 2022). Hence, we also consider the composition in customary Hill notation (Hill, 1900).

**SLICES** In addition to composition information, SLICES encompasses the composition and bonding of atoms within and across the unit cell. It is an invariant and invertible string representation without explicit information about the atom coordinates (Xiao et al., 2023). The representation is a single-line string starting with elemental symbols within the unit cell and followed by bond descriptions in the format `uvxyz`. Here, `u` and `v` represent node indices, while `xyz` denotes the direction of the unit cell necessary to establish each bond connection across the unit cell boundaries.

**CIF ($P_1$)** Crystallographic Information Files (CIFs) are a standard way to archive structural data in crystallography (Hall et al., 1991). They have been previously used for generating materials by fine-tuning LLMs (Flam-Shepherd & Aspuru-Guzik, 2023) or pretraining small GPT models (Antunes et al., 2023). In the CIF $P_1$ representation, the crystal structure is represented in the lowest symmetry ($P_1$ space group). This means that if there is any symmetry in the crystal structure, it is not explicitly defined.

**Crystal-text-LLM** This representation is a condensed version of the CIF, which includes only the parameters necessary for building the crystal structure(Gruver et al., 2023) (without additional syntax of the CIF). Given the lattice parameters of the unit cell, atom types, and their coordinates, the bulk

material structure can be represented as a listing of element symbols and coordinates separated by linebreaks that are prefixed by the list of lattice parameters (cell lengths and angles). Contrasting the CIF and the Crystal-text-LLM representations allows us to obtain insights into the importance of the compactness of representations.

### 3.1.2 NEW MATERIAL REPRESENTATIONS

**Atom Sequence**  To investigate the effect of the representation of compositional information, we explicitly list all the atoms present within the unit cell to eliminate any confusion that might arise from interpreting numbers as stoichiometric coefficients. Concretely, structures are represented by listing each atom symbol $n$ times to denote repetition within the unit cell structure. This representation is an intermediate representation between SLICES and composition in Hill notation, allowing to ablate the relevance of bonding information in SLICES.

**Atom Sequence++**  We incorporate lattice parameters sequentially into the Atom Sequence to ablate the effect of having unit cell dimensions.

**CIF symmetrized**  This CIF representation represents the asymmetric unit and list the symmetry operations that can be applied to fill the unit cell by generating all equivalent positions. It typically contains fewer lines describing atoms' positions than the CIF in $P_1$ but extra text describing the symmetry operations. Thus, this representation can contain more tokens for some structures than the $P_1$ variant (CIF $P_1$). This representation allows us to elucidate the importance of explicit symmetry information alongside positional information.

**Z-matrix**  The z-matrix is a representation widely used as input for quantum mechanical simulations of small molecules (but not materials). It leverages internal coordinates and is hence invariant with respect to translation or rotation. The internal coordinates used in a z-matrix are bond distances, angles, as well as dihedral angles. As all of these internal coordinates are defined with respect to neighboring atoms, the representation implicitly also encodes bonds. Here, we define the z-matrix based on the atoms within one unit cell.

**Local-Env**  We also report a new text representation inspired by the frequently used inductive bias of locality and Pauling's rule of parsimony, which states that local environments tend to be redundant (Pauling, 1960). To derive the local environments, we perform the coordination environment analysis reported by Waroquiers et al. (2020), derive Wyckoff labels using `spglib` (Togo & Tanaka, 2018), and SMILES using `openbabel` (O'Boyle et al., 2011). We prefix the representation using the spacegroup number and then list the Wyckoff label and SMILES separated by line breaks for each local environment.

### 3.2 DATA PREPARATION

For each representation, we provide standardized splits of 30k, 100k, 300k, and 2M samples of crystal structures in different text representations to enable researchers to study the effects of different data scales. To prepare the data, we first filtered out non-unique materials based on the NOMAD ID as a first step to address potential duplicates returned from API requests and then based on string matching of canonicalized CIF files from a set checkpoint of the NOMAD database. We also shuffled the dataset. For consistency, we limited the data to structures for which geometry was optimized with the PBE functional and the VASP electronic structure code. Using the shuffled dataset, we created a test holdout set (20k samples) by random sampling. Subsequently, we generated multiple subsets of varying sizes from the remaining data (30k, 100k, 300k, and 2M samples). Each subset was created by selecting the required number of examples sequentially. This method ensures that the smaller subsets are proper subsets of the larger ones.

For the analysis, we excluded datasets with 2D structures and with high standard deviation on the leaderboard from those originally reported in `MatBench`. However, our MatText package allows users to seamlessly also leverage the other `MatBench` tasks.

# 4 MATTEXT ANALYSIS

We conduct a comprehensive study using our MatText framework and expose significant shortcomings of existing text-based material modeling methods. Our experimental design closely resembles the standard pretraining of encoder-only transformers and fine-tuning of decoder-only transformers for materials science and chemistry (Trewartha et al., 2022; Rubungo et al., 2023; Schwaller et al., 2021). Our results show that: 1. scaling of pretraining does not necessarily lead to better performance on downstream property-prediction tasks (Section 4.2.1); 2. that locality is an important inductive bias, and, most importantly; 3. that current language modeling frameworks do not effectively leverage geometric information. To further verify this hypothesis, we conduct a study with a separable physic-inspired potential function in Section 4.3. The procedure allows us to precisely ablate the contributions of compositional and geometric features and further confirms the general results of our analysis that current language-modeling frameworks struggle to leverage geometric information and do not consistently improve by increasing pretraining scale.

Table 2: MatText Performance of models trained on different representations. The table compares the performance of different representations in predicting material properties using MatText-BERT with tokenizers that split numbers into individual digits and Regression Transformer-based number tokenization (Born & Manica, 2023), respectively. Additionally, we report the performance of MatText-Llama of different sizes and the performance of BERT in a classification task. Root mean squared error (RSME, lower is better ↓) is the performance measure for regression tasks and Receiver Operating Characteristic/Area Under the Curve (ROC AUC, higher is better ↑) for classification task; the reported error indicates the standard deviation across the cross-validation folds. The models listed here are the best among the data scaling experiments for their respective representations. The SOTA (State of the Art) refers to the best-known performance from the `MatBench` leaderboard (Dunn et al., 2020).

| Task | SLICES | CIF $P_1$ | Composition | CIF Symmetrized | Z-Matrix | Local-Env | SOTA[*] |
|---|---|---|---|---|---|---|---|
| **BERT Custom Number Tokenizer[**]** | | | | | | | |
| Shear Modulus (GPa) ↓ | **0.144±0.002** | 0.152±0.003 | 0.175±0.006 | 0.152±0.007 | 0.153±0.002 | 0.169±0.008 | 0.108±0.001[a] |
| Bulk Modulus (GPa) ↓ | **0.149±0.003** | 0.154±0.007 | 0.173±0.006 | 0.153±0.008 | 0.152±0.002 | 0.154±0.007 | 0.104±0.004[a] |
| Perovskites (eV) ↓ | 0.099±0.007 | 0.095±0.007 | 0.563±0.006 | 0.109±0.008 | **0.095±0.005** | 0.098±0.009 | 0.055±0.004[a] |
| Bandgap (eV) ↓ | 0.843±0.009 | 1.565±0.033 | 0.959±0.010 | 1.125±0.030 | 1.095±0.013 | **0.805±0.005** | 0.396±0.005[a] |
| Formation Energy (eV) ↓ | 0.349±0.017 | 0.681±0.009 | 0.407±0.008 | 0.358±0.020 | 0.826±0.017 | **0.255±0.007** | 0.048±0.06[a] |
| Is Metal (ROC AUC) ↑ | 0.931±0.002 | 0.920±0.005 | 0.929±0.002 | 0.932±0.002 | 0.875±0.004 | **0.934±0.004** | 0.952±0.007[b] |
| **BERT Regression Transformer Number Tokenizer[**]** | | | | | | | |
| Shear Modulus (GPa) ↓ | 0.150±0.006 | **0.136±0.004** | 0.174±0.005 | 0.153±0.002 | 0.154±0.005 | 0.146±0.002 | 0.108±0.001[a] |
| Bulk Modulus (GPa) ↓ | 0.144±0.003 | 0.143±0.004 | 0.173±0.007 | 0.156±0.006 | 0.154±0.006 | **0.143±0.003** | 0.104±0.004[a] |
| Perovskites (eV) ↓ | 0.101±0.005 | 0.098±0.007 | 0.561±0.007 | 0.108±0.010 | 0.097±0.002 | **0.096±0.005** | 0.055±0.004[a] |
| **Llama-3-8B-Instruct - Finetuned** | | | | | | | |
| Shear Modulus (GPa) ↓ | **0.288±0.076** | 0.343±0.130 | 0.365±0.152 | 0.342±0.132 | 0.382±0.038 | 0.329±0.071 | 0.108±0.001[a] |
| Bulk Modulus (GPa) ↓ | 0.460±0.246 | 0.315±0.077 | 0.302±0.108 | 0.402±0.182 | **0.219±0.017** | 0.480±0.004 | 0.104±0.004[a] |
| Perovskites (eV) ↓ | 0.294±0.150 | **0.181±0.018** | 0.716±0.033 | 0.225±0.031 | 0.286±0.045 | 0.410±0.045 | 0.055±0.004[a] |
| **Llama-2-7b-chat-hf - Finetuned** | | | | | | | |
| Shear Modulus (GPa) ↓ | 0.189±0.006 | 0.194±0.007 | 0.212±0.008 | 0.191±0.008 | 0.193±0.007 | **0.181±0.007** | 0.108±0.001 |
| Bulk Modulus (GPa) ↓ | 0.181±0.012 | 0.185±0.011 | 0.196±0.006 | 0.186±0.018 | 0.186±0.008 | **0.177±0.005** | 0.104±0.004 |
| Perovskites (eV) ↓ | 0.155±0.010 | **0.130±0.004** | 0.691±0.018 | 0.289±0.008 | 0.146±0.006 | 0.139±0.008 | 0.055±0.004 |
| **Llama-2-13b-chat-hf - Finetuned** | | | | | | | |
| Shear Modulus (GPa) ↓ | 0.179±0.007 | 0.180±0.005 | 0.210±0.007 | 0.184±0.009 | **0.178±0.007** | 0.183±0.009 | 0.108±0.001 |
| Bulk Modulus (GPa) ↓ | 0.168±0.006 | 0.172±0.009 | 0.197±0.013 | 0.181±0.010 | 0.173±0.009 | **0.163±0.012** | 0.104±0.004 |
| Perovskites (eV) ↓ | 0.133±0.005 | 0.227±0.035 | 0.689±0.016 | 0.206±0.014 | 0.146±0.006 | **0.125±0.004** | 0.055±0.004 |
| **Llama-2-70b-chat-hf - Finetuned** | | | | | | | |
| Shear Modulus (GPa) ↓ | **0.173±0.008** | 0.180±0.010 | 0.205±0.010 | 0.184±0.009 | 0.182±0.006 | 0.180±0.007 | 0.108±0.001 |
| Bulk Modulus (GPa) ↓ | 0.173±0.008 | **0.171±0.008** | 0.193±0.008 | 0.182±0.012 | 0.174±0.012 | 0.171±0.012 | 0.104±0.004 |
| Perovskites (eV) ↓ | 0.112±0.002 | 0.115±0.002 | 0.715±0.014 | 0.149±0.006 | 0.127±0.001 | **0.110±0.001** | 0.055±0.004 |

[a] connectivity-optimized nested GNNs Ruff et al. (2023).
[b] CGCNN v2019 Xie & Grossman (2018)
[*] Due to limitations in context length, some structures were filtered out during fine-tuning and testing to prevent ambiguity from truncated structures. More analysis justifying the filtering is provided in Appendix A.9. Note that SOTA results do not apply this filtering; hence, our metrics underestimate the performance of the text-based approaches.
[**] The "Custom Number Tokenizer" refers to the custom MatText tokenizers used for the respective representations without any special handling of numbers. The "Regression Transformer Tokenizer" applies special treatment for numbers in addition to the custom MatText tokenizers.

## 4.1 MATTEXT-LLAMA

Recent findings have shown that the fine-tuning of general-purpose (decoder-only) large language models can be an effective way to predict the properties of chemical compounds and materials (Jablonka et al., 2024; Rubungo et al., 2024). To understand the effect of materials representation in this setting, we performed parameter-efficient fine-tuning of two LLMs of different sizes, specifically Llama-3 8B model and Llama-2 13B model.

Interestingly, we find that these models struggle to leverage positional information (Table 2). In some cases, explicit positional information (e.g., CIF $P_1$ or Crystal-text-LLM) even detriments the performance of the model. A notable exception is the perovskites dataset, which has few unique chemical environments compared to the shear and bulk modulus datasets (see Appendix A.7), wherefore most of the variance cannot be explained using composition information alone. Importantly, however, the SLICES representation, which does not contain explicit geometric information, does not perform much worse than representations with explicit positional information (e.g., CIF), indicating that the models do not leverage explicit positional information.

While the increase in model size led to notable improvements in property prediction across all representations and properties, the observed lack of positional information utilization remains consistent (SLICES, Local-Env better than CIF in most of the cases). table 2. This suggests that while larger model sizes enhance overall predictive capability, they do not inherently address the specific challenge of leveraging positional information effectively in materials science applications.

## 4.2 MATTEXT-BERT

To determine if our findings are limited to the setting of fine-tuning of decoder-only LLMs, which have not been pretrained on materials representations, we pretrained BERT models (with 4 layers and 8 heads) using a masked language modeling objective for 50 epochs on our MatText representations. Following pretraining, we fine-tuned the models on the materials property prediction tasks.

### 4.2.1 PRETRAINING DATA SCALING

**The Limited Utility of Scaling Pretraining** In our pretraining scaling experiments (Figure 2), we find that the dataset size has little effect on the performance for some tasks (such as the prediction of the heat of formation of perovskites). This echos findings for protein language models that indicate that the scaling of pretraining does benefit many relevant downstream tasks (Li et al., 2024).

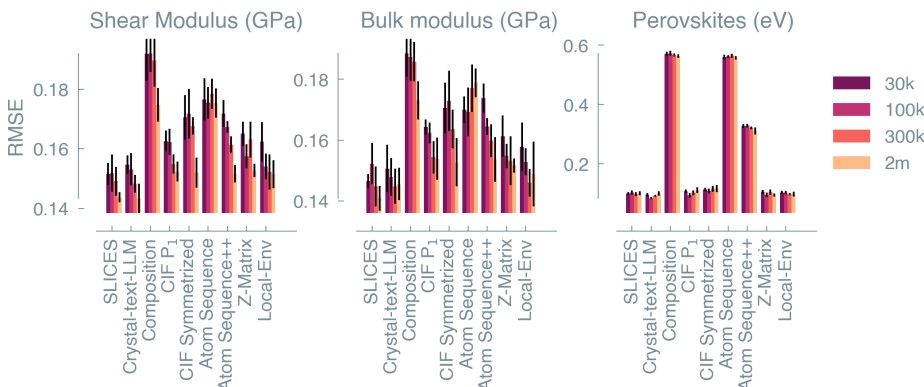

Figure 2: MatText-BERT scaling cross-validation scores. We pretrained BERT models with increasing pretraining dataset sizes (indicated in color) and fine-tuned them for material property prediction tasks using different material text representations. The error bars indicate the standard deviation across cross-validation folds.

**The Limited Utility of Geometric Information** However, the impact of the text representations is even more striking. Conventionally, one might expect that adding more information improves

predictive performance. For instance, one would expect that the addition of positional information increases performance in predicting the bulk and shear modulus (which one also observes for other models, e.g., GNNs, on these tasks). In our experiments, however, we find that this is not the case. In several instances, we could achieve comparable, if not better, performance with less information using a representation such as SLICES, which only contains information about the composition and the bond network. Notably, the predictive performance also does not improve when we incorporate classical inductive biases such as symmetry information (e.g., CIF Symmetrized) or internal coordinates (Z-Matrix). This indicates that the model cannot efficiently leverage the geometric information itself (encoded in numeric form) and supports our findings on MatText-Llama. Our attention analysis in Appendix A.6 shows further evidence on how language models place significant importance on compositional information and little on geometrical information. In addition, we also do not observe a noticeable impact from changing the tokenizer to one optimized for numbers (Born & Manica, 2023) in our pretraining and fine-tuning experiments (Appendix A.5), indicating that our findings are not specific to a particular choice of tokenizer.

**The Importance of Locality**   Importantly, our results show that other inductive biases can be more effective and, hence, can guide the development of better text representations. The good performance of SLICES seems to be based on the added bonding information in SLICES compared to Composition and Atom Sequence (Figure 2). Based on these findings and hypothesizing that models mostly learn by leveraging contributions of local environments (inspired by Pauling's redundancy rule (Pauling, 1960)), we developed a new representation, Local-Env, which is a text-based coarse-grained representation of the local environment. It does not contain explicit bonding and positional information but performs in many cases comparably to representations with explicit positional information or explicit bonding information while being more concise and readable.

### 4.3 HYPOTHETICAL POTENTIAL TO RELABEL STRUCTURES

A problem with many standard benchmarks is that they make it difficult to distinguish effects stemming from the data-generating process from those stemming from the model, which introduces some level of arbitrariness in the benchmarking results based on the choice of the dataset. For instance, positional features might only have a limited effect in our case because explicit geometric features could be unimportant for the task at hand. To remove this effect and to better understand the limitations of models in leveraging geometric information, we relabel crystal structures using a physics-inspired hypothetical potential, $E$, based on a composition- and a position-dependent term with independent generative processes.

$$E = \alpha E_{\text{comp}} + (1 - \alpha)E_{\text{pos}} = \underbrace{\sum_{k=1}^{k} w_k n_k}_{E_{\text{comp}}} + \underbrace{\sum_{i=1}^{N} V(\mathbf{r}_i) + \sum_{i=1}^{N} \sum_{j \in \mathcal{N}(i)} V(|\mathbf{r}_i - \mathbf{r}_j|)}_{E_{\text{pos}}}, \quad \alpha \in (0, 1)$$

(1)

where $w_k$ is a parameter associated with particles of type $k$. This term represents the contribution of each particle type's intrinsic properties to the system's total energy, $n_k$ is the number of particles of type $k$, and $\mathcal{N}(i)$ returns the neighborhood of $i$.

Using this formalism, we can easily measure the learning of positional information as well as long-range interactions by tuning these parameters in the parameters. For instance, when we tune the influence of geometric information by changing $\alpha$, we see that models fail to effectively capture positional information, with most representations showing a similar behavior as a function of $\alpha$ (Figure 3). For instance, for the structures of the GVRH and dielectric datasets, virtually all representations perform equally poorly in predicting the geometric component of the hypothetical energy. That is, independent of the presence or absence of geometric information, the model—in contrast to simple baseline models—fails to leverage it (even for our locality-constrained setup). Again, the perovskite dataset might appear as a special case at first glance. However, the representations with and without explicit geometric information perform equally well here, too. Notably, the very good performance of the representations with locality information in this task indicates that the model can learn shortcuts based on recurring local environment instead of learning the underlying physical relationship (Dziri et al., 2023; Geirhos et al., 2020; Zhou et al., 2023).

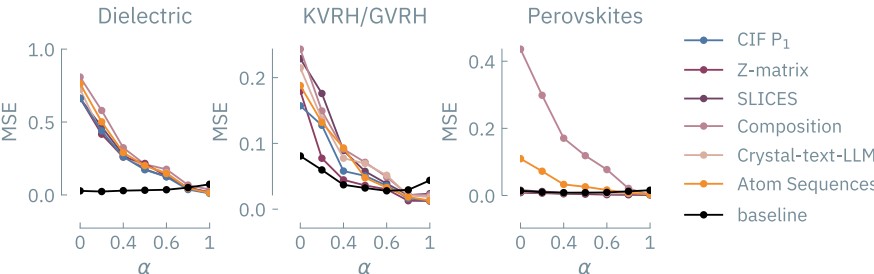

Figure 3: Performance of fine-tuned models as a function of the importance of geometric features. The $x$-axis represents the relative contribution ($\alpha$ in Equation (1)) of composition ($E_{\text{comp}}$) and position-dependent ($E_{\text{pos}}$) terms to the energy ($E$). The $y$-axis is the mean absolute error in predicting the hypothetical energy by our pretrained BERT (2M) models finetuned with different representations. As $\alpha$ increases, the geometric term ($E_{\text{pos}}$) becomes less dominant, leading to less error in predicting the energy. For this experiment, we chose a Lennard-Jones-like term for $V(\mathbf{r})$ in Equation (1). The KVRH and GVRH datasets contain the same structures but different labels. The baseline model refers to a gradient-boosting regression tree trained on a conventional material informatics descriptor (Appendix A.8) and shows less sensitivity with respect to $\alpha$.

Overall, our findings suggest that simply scaling pretraining may not improve performance on downstream tasks and that language modeling frameworks may not effectively utilize geometric data.

## 4.4 Limitations

The proximity of distributions of pretraining and downstream tasks is relevant for transfer performance (Hernandez et al., 2021; Cherti & Jitsev, 2022), and we cannot exclude the possibility that some downstream tasks are out-of-distribution. Additionally, other pretraining objectives (compared to Masked Language Modeling (MLM) or the general pretraining of a large language model) and model architectures might be better suited for our tasks. We also cannot exclude the potential for some emergent understanding if we were to increase the model scale, e.g., in the presence of grokking (Power et al., 2022).

## 5 Conclusions and Future Work

There has been considerable interest in modeling the properties of materials using language models. It is often believed that language models can solve almost any task, provided enough scale and data. Our findings indicate that for text-based materials modeling, more nuance is required. Specifically, our results demonstrate that additional information about the 3D geometry of materials, which one might expect to improve performance, does not help models and can sometimes even degrade performance. This observation holds true even when this data is seemingly enhanced using conventional inductive biases such as symmetry or internal coordinates. Consequently, our work suggests that conventional wisdom from building models on natural language may not be applicable to modeling materials using language models.

Moreover, our findings imply that, at least within the current language modeling frameworks (including both masked and causal language modeling), these models might not be the best solution for modeling materials. This mirrors observations such as the continued struggle of leading models with basic arithmetic tasks (Shen et al., 2023; Lee et al., 2023; Dziri et al., 2023). Zhou et al. (2023) proposed that transformers can generalize across task lengths if the task can be solved using a short so-called Restricted Access Sequence Processing Language (RASP) program (Weiss et al., 2021). Such RASP programs are intended to provide formal representation of programs that can be expressed using transformers and the the addition of numbers, for example, cannot be expressed with such a short RASP program. Modeling materials requires not only addition and subtraction but also more complex operations. From this perspective, our findings align with this conjecture and highlight significant limitations of current text-based modeling of material properties. This, however, must

not apply to the unconditional generation of materials, which might be represented using simpler programs and for which successes have been demonstrated using language models.

Positional encoding has been identified as one of the reasons for these issues (Nogueira et al., 2021). Therefore, our future work will leverage novel encoding schemes such as COPE (Golovneva et al., 2024), but also specific continuous number encoding techniques (Golkar et al., 2023).

Interestingly, our findings underscore that locality is one of the strongest inductive biases for materials modeling, suggesting that current language modeling setups might be most applicable where the property prediction problem can be solved using a group contribution formalism.

These findings highlight the importance and utility of our MatText framework. Practitioners can now easily create, use, and evaluate various text-based representations of materials in a systematic fashion. We believe that this framework will enable further systematic study of text-based materials modeling and thus expedite the design and discovery of novel materials.

## 6 ETHICS STATEMENT

The development and application of MatText, a suite of benchmarking tools and datasets for evaluating language models in materials science, have been conducted with careful consideration of ethical implications. However, we acknowledge the potential for the datasets and models in this work to be applied in the development of new materials and molecules. While the applications of this work hold significant potential, we acknowledge the possibility of dual use.

## 7 REPRODUCIBILITY STATEMENT

To facilitate the reproducibility of our work, we have provided all the computational details and hyperparameters in appendix A.3 and appendix A.4. We have uploaded the source code for pretraining, finetuning, analysis, and tooling to create all the representations and tokenization to `https://www.dropbox.com/scl/fo/afey2ymttiiaouzv31ld0/AKTTV3E9_k5rILd24FjiHBE?rlkey=3oncu3annj6yrsty6uq21naoy&dl=0`. All the checkpoints used in this work and the dataset can be downloaded with the same package (example code given in appendix A.2).

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

# A  APPENDIX

## A.1  MATTEXT REPRESENTATIONS

In Table 3, we list examples of the same material in various text representations.

Table 3: Example of MatText representations for one material.

| representation name | representation |
|---|---|
| Composition | `MgPtTa` |
| Atom Sequence | `Mg Ta Pt` |
| Atom Sequence ++ | `Mg Ta Pt 4.32 4.32 4.32 60 60 60` |
| SLICES | `Mg Ta Pt 0 2 - - o 0 2 - o - 0 2 - o o 0 2 o - - 0 2 o - o 0 2 o o - 0 1 - - o 0 1 - o - 0 1 o - - 1 2 o o o` |
| Crystal-text-LLM | `3.5 4.2 4.4`
`90 90 90`
`Ta`
`0.76 0.12 0.00`
`Ta`
`0.00 0.12 0.18`
`V`
`0.00 0.00 0.00`
`Ga`
`0.76 0.00 0.18` |
| CIF $P_1$ | `data_MgTaPt`
`_symmetry_space_group_name_H-M 'P 1'`
`_cell_length_a 4.32`
`_cell_length_b 4.32`
`_cell_length_c 4.32`
`_cell_angle_alpha 60.0`
`_cell_angle_beta 60.0`
`_cell_angle_gamma 60.0`
`_symmetry_Int_Tables_number 1`
`_chemical_formula_structural MgTaPt`
`_chemical_formula_sum 'Mg1 Ta1 Pt1'`
`_cell_volume 56.85`
`_cell_formula_units_Z 1`
`loop_`
`_symmetry_equiv_pos_site_id`
`_symmetry_equiv_pos_as_xyz`
`1 'x, y, z'`
`loop_`
`_atom_site_type_symbol`
`_atom_site_label`
`_atom_site_symmetry_multiplicity`
`_atom_site_fract_x`
`_atom_site_fract_y`
`_atom_site_fract_z`
`_atom_site_occupancy`
`Mg Mg0 1 0.0 0.0 0.0 1.0`
`Ta Ta2 1 0.58 0.58 0.58 1.0`
`Pt Pt1 1 0.53 0.53 0.53 1.0` |

| Z-Matrix | ```
Mg
Ta 1 6.1
Pt 2 0.5 1 0
``` |
| --- | --- |
| CIF-Symmetrized | ```
data_MgTaPt
_symmetry_space_group_name_H-M R3m
_cell_length_a 4.32
_cell_length_b 4.32
_cell_length_c 10.57
_cell_angle_alpha 90.0
_cell_angle_beta 90.0
_cell_angle_gamma 120.0
_symmetry_Int_Tables_number 160
_chemical_formula_structural MgTaPt
_chemical_formula_sum 'Mg3 Ta3 Pt3'
_cell_volume 170.55
_cell_formula_units_Z 3
loop_
_symmetry_equiv_pos_site_id
_symmetry_equiv_pos_as_xyz
1 'x, y, z'
2 '-y, x-y, z'
3 '-x+y, -x, z'
4 '-y, -x, z'
5 '-x+y, y, z'
6 'x, x-y, z'
7 'x+1/3, y+2/3, z+2/3'
8 '-y+1/3, x-y+2/3, z+2/3'
9 '-x+y+1/3, -x+2/3, z+2/3'
10 '-y+1/3, -x+2/3, z+2/3'
11 '-x+y+1/3, y+2/3, z+2/3'
12 'x+1/3, x-y+2/3, z+2/3'
13 'x+2/3, y+1/3, z+1/3'
14 '-y+2/3, x-y+1/3, z+1/3'
15 '-x+y+2/3, -x+1/3, z+1/3'
16 '-y+2/3, -x+1/3, z+1/3'
17 '-x+y+2/3, y+1/3, z+1/3'
18 'x+2/3, x-y+1/3, z+1/3'
loop_
_atom_site_type_symbol
_atom_site_label
_atom_site_symmetry_multiplicity
_atom_site_fract_x
_atom_site_fract_y
_atom_site_fract_z
_atom_site_occupancy
Mg Mg0 3 0.0 0.0 0.0 1.0
Ta Ta1 3 0.0 0.0 0.42 1.0
Pt Pt2 3 0.0 0.0 0.47 1.0
``` |
| Local Env | ```
R3m
Ta (1a) [Ta]#[Pt]
Pt (1a) [Ta]#[Pt]
Mg (1a) [Ta][Mg][Ta].[Ta].[Pt].[Pt].[Pt]
``` |

| Robocrys | MgTaPt crystallizes in the trigonal R3m space group. Mg(1) is bonded in a 6-coordinate geometry to three equivalent Ta(1) and three equivalent Pt(1) atoms. All Mg(1)-Ta(1) bond lengths are 2.66 Å. All Mg(1)-Pt(1) bond lengths are 3.22 Å. Ta(1) is bonded in a single-bond geometry to three equivalent Mg(1) and one Pt(1) atom. The Ta(1)-Pt(1) bond length is 0.55 Å. Pt(1) is bonded in a single-bond geometry to three equivalent Mg(1) and one Ta(1) atom. |
|---|---|

### A.2  MATTEXT PYTHON PACKAGE EXAMPLE USE CASES

All these representations can be easily obtained with the MatText Python package

```python
from mattext.representations import TextRep
from pymatgen.core import Structure

# Load structure from a CIF file
from_file = "InCuS2_p1.cif"
structure = Structure.from_file(from_file, "cif")

# Initialize TextRep Class
text_rep = TextRep.from_input(structure)

requested_reps = [
    "cif_p1",
    "slices",
    "atom_sequences_plusplus",
    "crystal_text_llm",
    "zmatrix"
]

# Get the requested text representations
requested_text_reps = text_rep.get_requested_text_reps(requested_reps)
```

**Custom MatText Tokenizer**    MatText provides tokenizers designed specifically for the MatText representations for more meaningful tokenization. All the MatText tokenizers have the option to enable special treatment for numbers first implemented in Born & Manica (2023).

```python
from mattext.tokenizer import SliceTokenizer

tokenizer = SliceTokenizer(
                model_max_length=512,
                truncation=True,
                padding="max_length",
                max_length=512
            )
tokenizer.tokenize("Ga Ga P P 0 3 - - o 0 2 - o - 0 1 o - -")

# output: ['[CLS]', 'Ga', 'Ga', 'P', 'P', '0', '3', '- - o', '0',
 '2', '- o -', '0', '1', 'o - -', '[SEP]']

tokenizer = SliceTokenizer(
                special_num_token=True,
                model_max_length=512,
```

```
                    special_tokens={},
                    truncation=True,
                    padding="max_length",
                    max_length=512
            )
tokenizer.tokenize("H2SO4")

# output: ['H', '_2_0_', 'S', 'O', '_4_0_']

More Examples

#Slice Tokenization: ['[CLS]', 'Ca', 'Hg', 'O', 'O', '0', '3',
'o o +', '0', '3', 'o + o', '0', '3', '+ o o', '0', '2', '- o o',
'0', '2', 'o - o', '0', '2', 'o o -', '1', '2', '- - -', '1', '3',
'o o o', '[SEP]']
#Slice Tokenization (Regression Transformer): ['[CLS]', 'Ca', 'Hg',
'O', 'O', '_0_0_', '_3_0_', 'o o +', '_0_0_', '_3_0_', 'o + o',
'_0_0_', '_3_0_', '+ o o', '_0_0_', '_2_0_', '- o o', '_0_0_',
'_2_0_', 'o - o', '_0_0_', '_2_0_', 'o o -', '_1_0_', '_2_0_', '- -
-', '_1_0_', '_3_0_', 'o o o', '[SEP]']

# Representation with locality encoded (local env)
from mattext.tokenizer import SliceTokenizer

local_env = "P1 Rb (1a) [Cs][As]([Cs])[Cs].[Cs][As][Cs].[As][Rb].[As][Cs] As (1a) [R
composition = "AsCsRb"

tokenizer = SmilesTokenizer (
    model_max_length =512 ,
    truncation =True ,
    padding =" max_length ",
    max_length =512,
    special_num_token=False ,
    )

#output:
#['[CLS]', 'P', '1', ' ', 'Rb', ' ', '(', '1', 'a', ')', ' ', '[', 'Cs', ']',
'[', 'As', ']', '(', '[', 'Cs', ']', ')', '[', 'Cs', ']', '.', '[', 'Cs', ']',
'[', 'As', ']', '[', 'Cs', ']', '.', '[', 'As', ']', '[', 'Rb', ']', '.', '[',
'As', ']', '[', 'Cs', ']', ' ', 'As', ' ', '(', '1', 'a', ')', ' ', '[', 'Rb',
']', '[', 'As', ']', '(', '[', 'Cs', ']', ')', '[', 'Rb', ']', '.', '[', 'Rb',
']', '[', 'Cs', ']', '.', '[', 'Rb', ']', '.', '[', 'Cs', ']', '.', '[', 'Cs',
']', ' ', 'Cs', ' ', '(', '1', 'a', ')', ' ', '[', 'Rb', ']', '[', 'As', ']',
'(', '[', 'Rb', ']', ')', '[', 'Rb', ']', '.', '[', 'Rb', ']', '[', 'As', ']',
'[', 'Rb', ']', '.', '[', 'As', ']', '[', 'Rb', ']', '.', '[', 'As', ']', '[',
'Cs', ']', '[SEP]']
```

**Input to models in LLM finetuning**

```
f"[INST] <<SYS>> What is the {self.property_} of {rep}
Answer: [/INST] {label:.3f} "

# Here property, representation and labels would be replaced during training.
```

### A.3 MATTEXT-BERT COMPUTATIONAL DETAILS

We used the BERT base-uncased model with a configuration of hidden size 512, 4 hidden layers, 8 attention heads Devlin et al. (2018), and absolute positional embedding with a maximum position embedding of 1024 as the architecture for the MatText-BERT model.

We choose a batch size and context length specific to representation and training models for a total of 50 training epochs and a learning rate of $2 \times 10^{-4}$ using a masked language modeling (MLM) approach with a probability of 0.15 for masking tokens. Due to limitations in available computing resources, we employ different batch sizes for models.

For fine-tuning, we employ early stopping with a patience of 10 and a threshold of 0.001 to prevent overfitting, utilizing 20% of the data for evaluation while the remaining 80% is used for training. The learning rate is set to $2 \times 10^{-4}$. The pretrained base model layers are not frozen, and a regressor head on top of the base model is used for the regression where the embedding of the first token (`[CLS]` token) is used as the feature.

The Pretraining and finetuning were conducted using NVIDIA A100 GPUs, with variants of 80 GB and 40 GB memory. Pretraining representations with a context length of 512 took approximately 8, 24, and around 48 h on a single A100 GPU for datasets of 30k, 100k, and 300k, respectively. Pretraining with 2 million data points was completed in 8 A100 GPUs with 40 GB memory each, totaling 30 h. For representations with a context length of 1024, the computation time ranged from 18 h to 3 days for datasets of 30k to 2 million points. Context length of 32 required significantly less time, approximately 40 min, 120 min, and 250 min for 30k, 100k, and 300k datasets, respectively, and around 8 h to 12 h for the 2 million dataset. Finetuning on perovskite structures took 2 h, 4 h to 5 h, and 8 h for context lengths of 32, 512, and 1024, respectively, while for KVRH and GVRH models, it took slightly more than half of the respective finetuning durations.

Table 4: Representations and their corresponding context lengths.

| Representation | Context Length |
|---|---|
| SLICES | 512 |
| Composition | 32 |
| Crystal-text-LLM | 512 |
| Z-Matrix | 512 |
| CIF P$_1$ | 1024 |
| CIF Symmetrized | 1024 |
| Atom Sequence | 32 |
| Atom Sequence++ | 32 |
| Local-Env | 512 |

### A.4 MATTEXT-LLAMA COMPUTATIONAL DETAILS

Following related work (Gruver et al., 2023; Rubungo et al., 2024) we finetune Llama-3-8B (Touvron et al., 2023) models with LoRA (Hu et al., 2021). In addition, we use 4-bit quantization with `nf4` quantization type and `float16` compute data type Dettmers et al. (2024). We employed a rank-size of 32 and $\alpha = 64$, with a batch size of 8 for 5 epochs and a cosine-annealed learning rate of 0.0003, with no bias applied and on a `CAUSAL_LM` task. We accumulated gradients over 4 steps and employed gradient checkpointing. The learning rate was set to $3 \times 10^{-4}$ with a cosine scheduler, a warmup ratio of 0.03, and a weight decay of 0.001. Optimization was performed using the `paged_adamw_32bit` optimizer. A maximum gradient norm of 0.3 was maintained to ensure stable training. On A100 80 GB GPUs, the finetuning and testing time varies from 120 min to 350 min depending upon the representation and finetune dataset size, with composition consuming the least time and CIF Symmetrized the most for the perovskite dataset. Packing during training was disabled, and a data collator was defined to train only for the generation part of the prompts. Fine-tuning LLAMA on 80 GB A100 GPUs for perovskites ranged from 3 h for smaller representations to 5 h (SLICES, Crystal-text-LLM) to 17 h (CIF Symmetrized), with other datasets generally requiring about half the compute time.

## A.5 TOKENIZATION INFLUENCE

One reason numbers might not be processed correctly is the tokenization method (Singh & Strouse, 2024). For instance, in many default tokenization methods (e.g., BPE, single-digit tokenization), numbers are represented with varying numbers of tokens, which might make it more difficult for models to use them effectively. To address this issue, we implemented the tokenizer proposed by Born & Manica (2023), which preserves decimal order and also encodes the order of magnitude. We find that this change does not lead to a noticeable impact on the results (Figure 4).

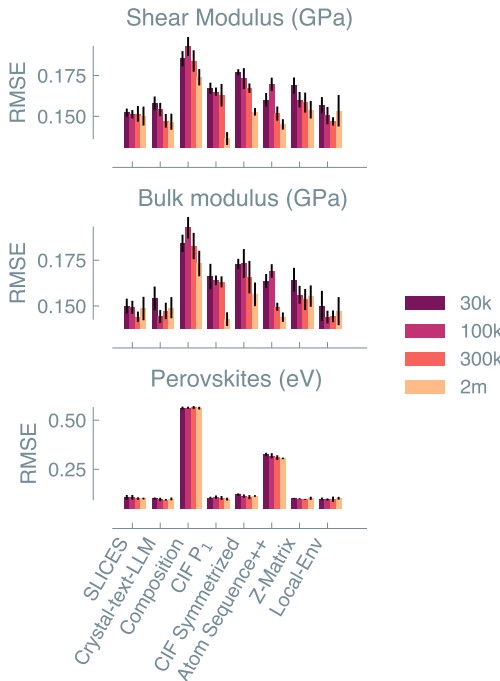

Figure 4: MatText-BERT pre-trained with increasing dataset sizes and fine-tuned for material property prediction using different material text representations, where numbers are tokenized as implemented in Regression Transformer (Born & Manica, 2023).

## A.6 ATTENTION ANALYSIS

To further elucidate why models seemingly falter to leverage explicit positional information encoded in numeric form, we analyzed attention weights. The amount of attention received by different tokens can be interpreted as a measure of the relevance of different tokens in the representation (Figure 2). The models here attend most to atomic symbols, which is consistent across all the representations and aligns with our other findings that composition/stoichiometry is an important feature contributing to accurate property prediction (Section 4.2.1). Consistently, we also observe that numbers receive less attention. Overall, this supports the hypothesis that current models do not effectively utilize numerical information for learning complex geometric features.

**Token attention contribution calculation** To perform this analysis, we first compute the contribution per token.

The element-wise multiplication of the attention matrix $A^{(l,h)}$ and mask $M_k$ gives the contribution of the attention scores for the token type $k$:

$$C_k^{(l,h)} = A^{(l,h)} \odot M_k$$

Here, $\odot$ denotes element-wise multiplication.

In this context, $A^{(l,h)}$ represents the attention matrix for layer $l$ and head $h$, and $M_k$ is the mask for token type $k$ in tokenized material text representations. Tokens can be classified into different types for analytical purposes. For example, the SLICES representations can have tokens of the type `ATOMS`, `NUMS`, and `DIR`. Specifically, all atoms are classified under the `ATOMS` token type, numbers are classified under `NUM`, and `DIR` represents tokens defining the direction of bonds.

The mask $M_k$ is defined as:

$$M_k \in \{0,1\}^{T \times T},$$

where $M_k$ is a binary matrix taking values 0 or 1.

The dimension of $M_k$ matches that of the attention weight matrix $A^{(l,h)}$. Given that samples in the dataset may contain varying numbers of atoms, each sample can have different corresponding masks. To facilitate this analysis, the MatText tokenizers provide the functionality to generate a list of token types alongside the list of tokens. These token types are used dynamically to design masks for attention analysis.

**Token Weight**   The percentage weight for token type $k$ in layer $l$ and head $h$ is then calculated as:

$$W_k^{(l,h)} = \frac{\sum_{i,j}\left(C_k^{(l,h)}(i,j)\right)}{\sum_{i,j} M_k(i,j)}.$$

Here $W_k^{(l,h)}$ is the percentage attention recieved by a particular token type $k$ in layer $l$ and head $h$ during prediction and $\sum_{i,j}$ denotes summing over all elements in the matrix.

**Aggregation Across Folds**   Aggregates the attention weights for all the samples across multiple folds. This involves summing the weights for each token across all samples ($N$) and folds ($f$). The total contribution of token type $k$ across all folds is given by:

$$T_k = \sum_N \sum_f \sum_{l,h} W_k^{(l,h,f)}.$$

**Results**   In the attention heat maps, we observe certain heads specialized to learn features from compositions, which is not the case for numbers, where we observe rather a dispersed nature in heads (Figure 6). Previously, with unsupervised language modeling of proteins, the formation of such heads was associated with parts of the architecture concentrating on learning certain features (Vig et al., 2021). We observe that groups dedicated to learning numerical features do not emerge with pretraining.

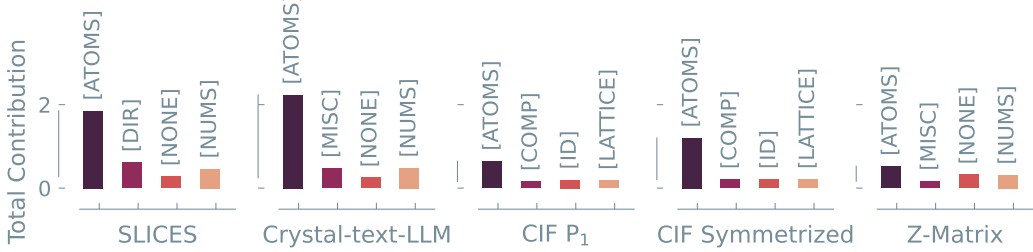

Figure 5: Attention received by different types of tokens in different representations summed across all the heads and layers

A.7   DIVERSITY ANALYSIS

To understand the impact of the dataset on the benchmark, we benchmarked the diversity of the datasets by computing the uniqueness and balance of local environments. To do so, we leverage the local environment analysis implemented for our Local-Env representations.

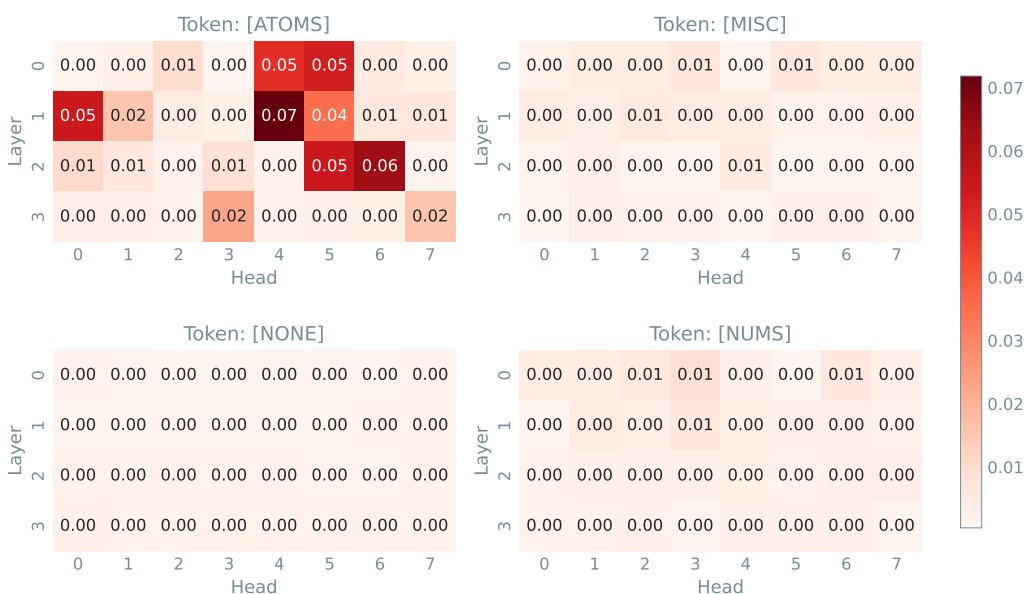

Figure 6: Attention contributions for different types of tokens in different heads and layers.

UNIQUENESS

$$\text{Uniqueness} = \frac{\{e\}}{\|e\|}, \tag{2}$$

where $e$ is a concatenation of the local environments of all structures.

BALANCE    Let $\mathbf{p} = \{p_1, p_2, \ldots, p_n\}$ be the probability distribution derived from the counter values, where $p_i = \frac{c_i}{\sum_{j=1}^{n} c_j}$ and $c_i$ are the counts of local environments.

$$H(\mathbf{p}) = -\sum_{i=1}^{n} p_i \log_2(p_i)$$

where $p_i \log_2(p_i)$ is defined to be 0 if $p_i = 0$.

We define balance as the entropy normalized by the maximum entropy:

$$\text{Balance} = H_{\text{norm}} = \frac{H(\mathbf{p})}{H_{\text{max}}} = \frac{-\sum_{i=1}^{n} p_i \log_2(p_i)}{\log_2(n)} \tag{3}$$

Table 5: Diversity analysis of the fine-tuning datasets. We compute diversity metrics based on local environment features (i.e., SMILES derived for coordination environments). The perovskite dataset has the lowest number of unique structures.

| dataset | number of structures | local env count | uniqueness | balance |
|---|---|---|---|---|
| Shear Modulus (GPa) | 38025 | 12293 | 0.11 | 0.92 |
| Bulk Modulus (GPa) | 38025 | 12293 | 0.11 | 0.92 |
| Perovskites (eV) | 144660 | 13063 | 0.02 | 0.84 |
| Dielectric | 18190 | 7134 | 0.08 | 0.88 |

The low values for uniqueness indicate a high value for redundancy, supporting intuition based on Pauling's rule of parsimony (Pauling, 1929).

### A.8 MODELING THE HYPOTHETICAL POTENTIAL USING "CONVENTIONAL" MODELS

To test whether the behavior of the test error as a function of $\alpha$ in Equation (1) is specific to the language modeling setup, we also trained models that mimic established setups in material informatics. We use `Dscribe` (Himanen et al., 2020) to derive Many-Body Tensor Representation features (Huo & Rupp, 2022) (using the pairwise inverse distance as geometry function on a grid with 250 points between 0 and 7, smoothed with a Gaussian of width 0.1 and exponential weighting) to describe the geometry of the structures and featurize the composition using `matminer` Ward et al. (2018), where employ the fraction of element, the mass of the atoms in the unit cell, the number of atoms in the unit cell, $p$-norms of stochiometric attributes (Ward et al., 2016). We then train a Histogram-based Gradient Boosting Regression Tree (Pedregosa et al., 2011) to predict the energy based on the features.

### A.9 FILTERING TRUNCATED STRUCTURES

One limitation of language modeling is the fixed sequence lengths the model can take as input. In order to deal with this, we modeled the longer representations with bigger context lengths (for example, a context length of 1024 for CIF-based representations). However, incorporating all the structures in the `MatBench` is impractical as it would demand a context length of more than 2048 (around 4 times higher computational cost). We hence created a separate filtered version of `MatBench` dataset where structures with sequence lengths beyond their respective context length when represented as CIF ($P_1$), CIF Symmetrized, SLICES, or Crystal-text-LLM are filtered out. However, this filtering does not make the comparisons erroneous but removes all the ambiguity that can arise from having truncated and nonphysical structures in the analysis. This is also supported by our further analysis of prediction error correlation with sequence length, where we do not notice any correlation that can arguably suggest the model performs poorly on larger sequence material. In fact, this filtering step can make the task more difficult as there are fewer samples in the training set. Therefore, we suggest using the MatText-filtered dataset for researchers limited by compute and MatText-matbench otherwise. Figure 7 shows the prediction error for samples with different sequence lengths. Perovskite structures are relatively small (five atoms in a unit cell) and do not undergo filtering.

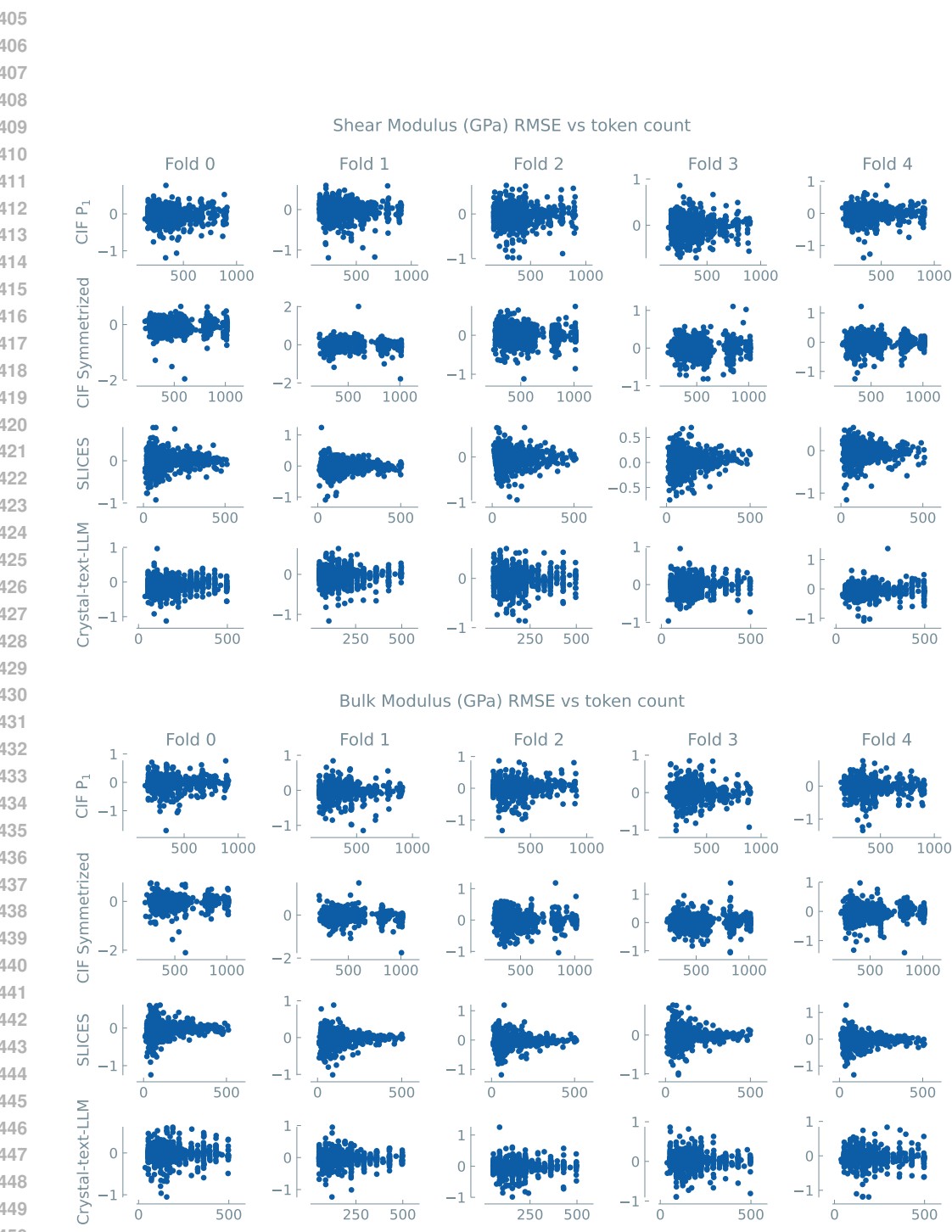

Figure 7: Prediction error correlation with sequence length for Shear Modulus and Bulk Modulus datasets.

