# OpenReview forum: "MatText: Do Language Models Need More than Text & Scale for Materials Modeling?"
_ICLR.cc/2025/Conference — Submitted to ICLR 2025_

### Official Review · Reviewer_q6NL · 2024-10-28

**Soundness:** 3
**Presentation:** 2
**Contribution:** 2
**Rating:** 3
**Confidence:** 2

**Summary:**

This paper proposes a benchmark and toolkit for evaluating language models for modeling materials. By evaluating current LLMs on different text representations of materials from data sizes and inductive biases, the authors find problems in modeling materials with LLMs.

**Strengths:**

- Several experiments from different aspects are provided such as different text representations, different data sizes, and LLaMa and BERT.
- Many recent methods are surveyed and summarized, and tasks, datasets, and toolkits are provided.

**Weaknesses:**

- The comparison and results are meaningful for materials LLM community, but the technical novelty is weak.
- The details of tasks and results are not well explained, and only numerical scores are shown, so it is difficult for readers unfamiliar with the material's domain to follow the paper.
- The claims on the limitations of LLMs are too strong considering the grokking or emergent abilities of LLMs.
  - Wei et al., Emergent Abilities of Large Language Models, TMLR, 2022

**Questions:**

See the weaknesses above.

---

> ### Author Response · Authors · 2024-11-26
> **Response to reviewer q6NL**
>
> > The comparison and results are meaningful for materials LLM community, but the technical novelty is weak.
> The details of tasks and results are not well explained, and only numerical scores are shown, so it is difficult for readers unfamiliar with the material's domain to follow the paper.
> The claims on the limitations of LLMs are too strong considering the grokking or emergent abilities of LLMs.
> Wei et al., Emergent Abilities of Large Language Models, TMLR, 2022
>
>
>
> While we do not propose new model architectures, our work's novelty lies in its systematic and comprehensive analysis of language models' capabilities in materials science. Our technical contributions of
>
> - Providing a Python package with consistent implementation of materials representations and tokenizers and interface to datasets
> - Conducting the first comprehensive benchmark (almost 1k training runs) in different language modeling paradigms (causal and masked), with different tokenizers and different text representations across multiple scales (from 7B to 70B parameter LLama-2
> - Proposing a novel analysis procedure (via the hypothetical potential, Section 4.3) that provides clear, quantitative evidence of LLMs' limitations in processing geometric information
>
> Led to important novel scientific insights
>
> (1) *More information is not better.* Geometric information is not effectively leveraged but the currently predominately used text-based modeling approaches for materials.
> (2) *More scale is not better.* In the current regimes (for pretraining we used one of the largest existing materials datasets) there are only marginal benefits of scaling.
>
> To reflect those insights we also changed our paper title from a question to a statement: "Language Models Need More than Text \& Scale for Materials Modeling"
>
> While we acknowledge the phenomenon of grokking and emergent abilities in LLMs (discussed in Section 4.4), our experiments suggest that the fundamental limitations we observe persist even with significant scaling. To make our results more accessible to readers outside the materials domain, we have included detailed examples of different representations in Table 3 and comprehensive analysis in Section 4.

---

### Official Review · Reviewer_nB8q · 2024-11-04

**Soundness:** 2
**Presentation:** 3
**Contribution:** 2
**Rating:** 5
**Confidence:** 5

**Summary:**

The paper studies the potential of using LLMs for materials science, specifically in materials modeling. The authors introduce MatText, a benchmarking suite with nine text-based representations tailored for material systems, incorporating unique inductive biases to capture physical properties. The different analysis using MatText reveals that current LLMs struggle to effectively leverage geometric information, which is crucial in materials science, and instead rely heavily on local information. This suggests that text-based LLM approaches, unlike in natural language tasks, may not be inherently suitable for capturing the complexities of material properties.

**Strengths:**

-  MatText includes different text-based representations and standardized datasets for evaluating LLMs on materials science tasks.
- The study identifies critical challenges, such as the inability of LLMs to leverage geometric information
- Provides tools for converting crystal structures into text and standardizing materials datasets.

**Weaknesses:**

- While the authors observe that LLMs focus on local information rather than spatial relationships (which have been noticed already in similar studies), they don’t provide a mechanism or architectural change to address this.

- The authors introduce nine different text-based representations of materials, each with unique inductive biases. However, text representations may be an inherently limited approach for encoding complex 3D geometric and physical properties. Certain representations (e.g., CIF) may add complexity without necessarily improving model performance, as the LLMs struggle with numerical encoding of positional data. In my opinion evaluating LLM on handling such representations is not fair.

- The experiments is conducted with different tokenizers for handling numerical information. However, standard tokenization methods for numbers can inpuct the model’s ability to process continuous data, such as atomic positions or lattice parameters. This point needs further clarification.

**Questions:**

Does the Local-Env representation, by focusing primarily on local information, risk overlooking other crucial features such as long-range interactions or periodicity? If so, how can we address this limitation?

---

> ### Author Response · Authors · 2024-11-26
> **Response to Reviewer nB8q**
>
> > While the authors observe that LLMs focus on local information rather than spatial relationships (which have been noticed already in similar studies), they don’t provide a mechanism or architectural change to address this.
>
> We thank the reviewer for highlighting one of the main findings of our work.
> The fact that (L)LMs do not effectively leverage geometric information has not been shown in materials and chemistry science and our work shows that it can neither be overcome by changing representation, modeling architecture, realistically achievable model and dataset set, and tokenization.
>
> Our work provides novel, quantitative evidence through the innovative hypothetical potential experiments (Section 4.3) that precisely ablates compositional and geometric contributions.
> This controlled experimental setup allows us to systematically measure how models leverage different types of information, going beyond just observing the focus on local features.
> Our attention analysis (Appendix A.6) provides detailed insights into how models process different types of information, showing that atomic symbols receive significantly more attention than geometric data across all representations. These insights are crucial for the development of more effective approaches to materials modeling using language models. and  provide deeper insights into how language models process and utilize materials information - insights that can guide the development of more effective approaches.
>
>
> > The authors introduce nine different text-based representations of materials, each with unique inductive biases. However, text representations may be an inherently limited approach for encoding complex 3D geometric and physical properties. Certain representations (e.g., CIF) may add complexity without necessarily improving model performance, as the LLMs struggle with numerical encoding of positional data. In my opinion evaluating LLM on handling such representations is not fair.
>
> We share the reviewer's skepticism. Similar concerns about the increasing number of reports using LLMs on representations like CIF or XYZ (e.g.  Gruver et al., Flam-Shepherd  & Aspuru-Guzik, Jacobs et al., Antunes et al.) motivated us to construct this benchmark.
>
> Our goal was to go deeper than previous work on finetuning LLMs on chemical or materials science text and understand where the limits are to enable future work to improve representations and/or architectures.
>
> > The experiments is conducted with different tokenizers for handling numerical information. However, standard tokenization methods for numbers can inpuct the model’s ability to process continuous data, such as atomic positions or lattice parameters. This point needs further clarification.
>
> Also here our motivation is similar to the concern the reviewer raised. Since for geometric information the representation of numbers is of utmost performance we investigated different tokenization approaches (Table 2).  A key finding we obtained using our custom number tokenization implementations is that the limitations persist across different numerical encoding schemes. Thus, those limitations are unlikely to be overcome with minor technical interventions.
>
> > Does the Local-Env representation, by focusing primarily on local information, risk overlooking other crucial features such as long-range interactions or periodicity? If so, how can we address this limitation?
>
> A crucial finding of our work is that Local-Env does not perform, sometimes even better, than representations that have this additional information. This shows that additional information, in the current approaches, is not always better.
>
> [Nate Gruver, Anuroop Sriram, Andrea Madotto, Andrew Gordon Wilson, C. Lawrence Zitnick, & Ulissi, Z. W. (2024). Fine-Tuned Language Models Generate Stable Inorganic Materials as Text. In International Conference on Learning Representations 2024.](https://arxiv.org/abs/2402.04379)
>
> [Flam-Shepherd, D., & Aspuru-Guzik, A. (2023, May 9). Language models can generate molecules, materials, and protein binding sites directly in three dimensions as XYZ, CIF, and PDB files. arXiv. ](http://arxiv.org/abs/2305.05708)
>
> [Jacobs, R., Polak, M. P., Schultz, L. E., Mahdavi, H., Honavar, V., & Morgan, D. (2024, September 9). Regression with Large Language Models for Materials and Molecular Property Prediction. arXiv. ](http://arxiv.org/abs/2409.06080)
>
> [Antunes, L. M., Butler, K. T., & Grau-Crespo, R. (2024, February 12). Crystal Structure Generation with Autoregressive Large Language Modeling. arXiv.](http://arxiv.org/abs/2307.04340)

---

### Official Review · Reviewer_CyJR · 2024-11-06

**Soundness:** 3
**Presentation:** 4
**Contribution:** 2
**Rating:** 5
**Confidence:** 4

**Summary:**

This paper introduces a benchmark for evaluating language models at predicting various properties of a material given its composition and structural information. To test the ability of LMs at utilizing different types of information (e.g., composition vs geometry), the benchmark provides several text representations of a material. Further, standardized data splits at different data scales are provided to test scaling behaviors.

Results suggest that LLMs, both existing ones and those pretrained on materials data from scratch, struggle at utilizing geometric information when making predictions. Further scaling the pretraining data seems to provide limited benefits.

**Strengths:**

- Standardizing materials modeling with a benchmark providing precomputed representations and data splits is very useful. This data could  be used in a lot of follow up research.
- Evaluations point out an interesting limitation of LLMs at utilizing geometric information effectively.

**Weaknesses:**

- One of the main claims of the paper is that scaling pretraining alone may not be sufficient for improving materials modeling performance. However, this is based only on increasing data size (which still shows improvement on 2 out of the 3 tasks). However, existing studies of scaling laws all note that all three -- model size, training compute and pretraining data size -- must be scaled in tandem for optimal performance. Hence, it is not clear if the observations on scaling would still hold if we increase model size and train for longer.

- The contributions of the paper are a bit thin -- the data itself is sourced from existing resources (NOMAD and MatBench). The main contribution are the different text representations evaluated on MatBench tasks. None of these lead to an improvement over existing methods.

- Only a small set of materials modeling tasks were evaluated -- so while the benchmark is somewhat comprehensive in the representations it uses, it is not comprehensive in the modeling abilities it tests.

**Questions:**

- Why "datasets with 2D structures were excluded" from MatBench?

---

> ### Author Response · Authors · 2024-11-26
> **Response to Reviewer CyJR**
>
> > One of the main claims of the paper is that scaling pretraining alone may not be sufficient for improving materials modeling performance. However, this is based only on increasing data size (which still shows improvement on 2 out of the 3 tasks). However, existing studies of scaling laws all note that all three -- model size, training compute and pretraining data size -- must be scaled in tandem for optimal performance. Hence, it is not clear if the observations on scaling would still hold if we increase model size and train for longer.
>
> We thank the reviewer for acknowledging a core contribution of our work: highlighting the limitations of LLMs in effectively utilizing geometric information - such that even scaling only yields marginal benefits.
>
> We would like to address each point:
>
> 1. We have now conducted extensive scaling experiments across model sizes (7B, 13B, and 70B parameters) and found no meaningful improvement in performance, even with these significantly larger models.
> This limitation persists across:
>    - Different representations
>    - Different tasks
>    - Different modeling paradigms (masked and causal)
>
> This new evidence strongly supports our original observation that simply scaling current language modeling approaches may be insufficient for materials modeling.
>
> 2. While we acknowledge that optimal scaling requires balancing model size, compute, and data, we face two fundamental constraints in materials science: a) The amount of available validated materials structure data is inherently limited, unlike web text b) The extreme compute requirements for optimal scaling would be impractical for most materials research
>
> > The contributions of the paper are a bit thin -- the data itself is sourced from existing resources (NOMAD and MatBench). The main contribution are the different text representations evaluated on MatBench tasks. None of these lead to an improvement over existing methods.
>
> The fact that we could not see an improvement is one of the main contributions of our work. In our work, we laid the foundation for further studies by
>
> - Providing a Python package with consistent implementation of materials representations and tokenizers and interface to datasets
> - Conducting the first comprehensive benchmark (almost 1k training runs) in different language modeling paradigms (causal and masked), with different tokenizers and different text representations
> - Proposing a novel analysis procedure (via the hypothetical potential)
>
> Those important technical advancements led us to already extract scientific insights:
>
> (1) *More information is not better.* Geometric information is not effectively leveraged but the currently predominately used text-based modeling approaches for materials.
>
> (2) *More scale is not better.* In the current regimes (for pretraining we used one of the largest existing materials datasets) there are only marginal benefits of scaling.
>
>
> > Only a small set of materials modeling tasks were evaluated -- so while the benchmark is somewhat comprehensive in the representations it uses, it is not comprehensive in the modeling abilities it tests.
>
> While we focused on a selected set of tasks, our analysis is still comprehensive:
>
> - Our ~1k training runs provide robust statistical evidence
> - The selected properties show different dependencies on geometric information
> - Our Python implementation allows easy extension to additional tasks
> - For comparison, recent related work often focuses on single properties (e.g., Li et al. 2024)
>
> > Why "datasets with 2D structures were excluded" from MatBench?
>
> The reason for not including 2-D structures is the fact that we only included 3D structures in the BERT pretraining.
>
> [Li, K., Rubungo, A. N., Lei, X., Persaud, D., Choudhary, K., DeCost, B., et al. (2024, June 10). Probing out-of-distribution generalization in machine learning for materials. arXiv.](http://arxiv.org/abs/2406.06489)

---

### Official Review · Reviewer_VmTz · 2024-11-06

**Soundness:** 2
**Presentation:** 3
**Contribution:** 2
**Rating:** 3
**Confidence:** 5

**Summary:**

In this paper the authors propose a benchmarking suite for evaluating LLMs in modeling materials, which includes evaluations as well as tools for converting materials to LLM-appropriate representations and calculating materials-specific metrics, such as sensitivity to crystal structure. The benchmark includes 9 distinct representations for encoding materials (some of which are newly proposed in the paper), with the shared task across representations of predicting material properties (e.g. shear modulus, bulk modulus, formation energy) given the input material representation. They use the benchmark to evaluate BERT pretraining and Llama (parameter efficient) finetuning (Llama-3-8B-Instruct, Llama-2-13b-chat-hf) on this task, comparing to the current state-of-the-art model according to the MatBench benchmark (variants of graph NNs). In their experiments they find that scaling pretraining (in a small BERT model) does not seem to improve performance on the benchmark, that locality (encoded into one of the input representations) is a useful inductive bias, and that current models do not seem to use geometric information.

**Strengths:**

- **Important and interesting area of research.** Leveraging LLMs for reasoning in the space of material composition, structure, and properties is an exciting and potentially impactful area of research, and developing benchmarks and software libraries towards this end should encourage further developments in this space. The authors are spot on in terms of the key challenge in this space, which is how to effectively represent rich metadata about materials composition and structure.
- **Clearly written and easy to understand.** The paper was generally well written and easy to follow.

**Weaknesses:**

- **Lack of engagement with prior work exploring methods for encoding structured/non-text information in language models (discussion or implementation).** The focus of the work / main findings are that LLMs are not very good at encoding/reasoning over numerical/structured values such as those important in representing and reasoning over the relationship between material composition and properties. But as far as I can tell, the paper only experiments with the most naive possible input and output representations (input: encoded representation of material composition/structure; output: numerical value of property). In this scenario, it is expected that the model will not do well in leveraging encoded structure because it is essentially forced to learn from scratch, because the representation of structure being used during finetuning/testing is likely too distinct from the representation of any similar knowledge during training, where the model was largely exposed to natural language text and code. It is known that LLMs, particularly the instruction-fine tuned models used in the paper, can struggle when provided with this type of non-natural-language formatted data, due to the mismatch between distributions observed during training, finetuning and evaluation. See e.g. [Fatemi et al. (2023)](https://arxiv.org/pdf/2310.04560), [Hegselmann et al. 2023](https://proceedings.mlr.press/v206/hegselmann23a.html), [Li et al. 2024](https://arxiv.org/abs/2403.07969), [Yin et al. 2023](https://arxiv.org/pdf/2306.01150), and other referenced work. While it would be interesting to validate this in this domain, insights from the current experimentation are limited because the experiments do not represent a realistic, good-faith setup for probing the models’ capabilities towards this end. Example methodologies that might do better would be tool use (to outsource quantitative reasoning), encoding the material structure information into code-type snippets, or even just encoding the input-output pairs as natural language questions (the Robocrys encoding gets close to this by at least encoding materials as natural language descriptions.) A single specific approach to this, RASP (Weiss et al. 2021), is discussed in the conclusions section, but not experimented with.
- **Limited contributions over prior work.** Relationship with MatBench should be clearly characterized very early in the paper, due to the similarity. MatBench is designed to benchmark any type of model on the same task, and the main contributions of this work over MatBench are: additional input representations (since this is the main challenge in using LLMs for this type of modeling) and software to help with evaluation and converting between representation formats. Because of the limited empirical results (as discussed above), this paper mostly provides a new benchmark, which is mostly composed of existing data/representations, but not many insights based on the benchmark, or other findings demonstrating its usefulness.
- **Limited analysis of results.** The appendix contains the beginning of some interesting analyses of the results. In particular, I would recommend moving discussion of the experimentation with tokenization to the main paper in future drafts, and elaborating. I’m very surprised that changes to the tokenization did not meaningfully improve performance. It would be interesting to see some examples of the tokenized inputs, under the default and modified settings. The encoded representations are so different than the text the default tokenizers were trained on, I would expect this to have a significant impact, and I wonder if just implementing Born & Manica (2023) was insufficient.
- **Discussion of limitations could be improved.** This work has many limitations not included. Breadth of models compared was limited. For example, benefits of scaling are only discussed in the context of “grokking” but it’s not clear why scaling more would not help further, since it was shown to lead to improvements in the existing results. The BERT models are incredibly small (4 layers and 8 attention heads), etc.

**Questions:**

- If you experimented with different prompt formats and strategies for decoder-only LLMs, could you report what you tried and whether it had any impact on results?
- If not, why did you use instruction fine-tuned models, but did not fine-tune on or experiment with instruction-style input-output pairs?
- Why not experiment with larger models (potentially proprietary/closed)?
- Why did you choose the specific size of the BERT model?

Additional notes:
- Would be helpful to add some additional highlighting in Table 2 — e.g. which representation (or MatBench model) performs best for each task? Very hard to parse visually. Also, both RMSE and ROC are reported, would be nice if it was indicated for which rows higher is better and lower is better.
- Fix references to be parenthetical (e.g. \citep rather than \citet) on line 84, also various similar mistakes in the appendix.
- Typo on line 332
- It would be helpful if you added some concrete examples of input-output pairs to the main paper, perhaps including examples that do and do not encode locality, since a distinction there was one of your key findings. Including versions tokenized using the given tokenizers would be informative as well.
- Relevant concurrent work: https://arxiv.org/abs/2409.14572

---

> ### Author Response · Authors · 2024-11-26
> **Rebuttal on setup not being a "not a realistic, good-faith setup"**
>
> We appreciate the thoughtful feedback. We must clarify several important points:
>
> > But as far as I can tell, the paper only experiments with the most naive possible input and output representations (input: encoded representation of material composition/structure; output: numerical value of property). In this scenario, it is expected that the model will not do well in leveraging encoded structure because it is essentially forced to learn from scratch, because the representation of structure being used during finetuning/testing is likely too distinct from the representation of any similar knowledge during training, where the model was largely exposed to natural language text and code. It is known that LLMs, particularly the instruction-fine tuned models used in the paper, can struggle when provided with this type of non-natural-language formatted data, due to the mismatch between distributions observed during training, finetuning and evaluation. See e.g. Fatemi et al. (2023), Hegselmann et al. 2023, Li et al. 2024, Yin et al. 2023, and other referenced work. While it would be interesting to validate this in this domain, insights from the current experimentation are limited because the experiments do not represent a realistic, good-faith setup for probing the models’ capabilities towards this end. Example methodologies that might do better would be tool use (to outsource quantitative reasoning), encoding the material structure information into code-type snippets, or even just encoding the input-output pairs as natural language questions (the Robocrys encoding gets close to this by at least encoding materials as natural language descriptions.) A single specific approach to this, RASP (Weiss et al. 2021), is discussed in the conclusions section, but not experimented with.
>
> We must add context to this claim.
>
> Some of our models have seen the text representations of materials in pretraining (thus, "representation [...] during finetuning/testing is likely too distinct from the representation [...] during training" is not precise). In our paper, we explore both the fine-tuning of models pre-trained on natural text (Llama family) and also BERT models that we pre-trained on the materials representations.
> This reflects that in chemistry and materials science, pretraining of BERT models (e.g.,  Chaudhari et al.) and finetuning of LLMs is popular and has seen some success (e.g., Jablonka et al., Van Herck et al., Gruver et al., Jacobs et al.)
>
> In particular, the fine-tuning of LLMs on text representations - such as the CIF-based representations we have used in our work - has been reported before at ICLR 2024 (Gruver et al.).
>
> However, while various works explored different representations -- ranging from compositions over robocrystallographer, to entire CIF files --- there is no real understanding of what makes for a good representation. Understanding this was the goal of our paper and the motivation for systematically implementing a software library for representations and running 405 Llama and 540 BERT finetune runs as well as 36 backbone training runs.
>
> Our results lead to the non-trivial insight that additional, physically meaningful information, such as positional information, cannot be effectively used by currently used approaches.
>
> [Nate Gruver, Anuroop Sriram, Andrea Madotto, Andrew Gordon Wilson, C. Lawrence Zitnick, & Ulissi, Z. W. (2024). Fine-Tuned Language Models Generate Stable Inorganic Materials as Text. In International Conference on Learning Representations 2024.](https://arxiv.org/abs/2402.04379) -
>
> [Chaudhari, A., Guntuboina, C., Huang, H., & Farimani, A. B. (2024, March 28). AlloyBERT: Alloy Property Prediction with Large Language Models. arXiv. ](http://arxiv.org/abs/2403.19783)
>
> [Jablonka, K. M., Schwaller, P., Ortega-Guerrero, A., & Smit, B. (2024). Leveraging large language models for predictive chemistry. Nature Machine Intelligence, 6(2), 161–169. ](https://doi.org/10.1038/s42256-023-00788-1)
>
> [Van Herck, J., Gil, M. V., Jablonka, K. M., Abrudan, A., Anker, A. S., Asgari, M., et al. (2024). Assessment of Fine-Tuned Large Language Models for Real-World Chemistry and Material Science Applications. Chemical Science.](https://doi.org/10.1039/D4SC04401K)
>
> [Jacobs, R., Polak, M. P., Schultz, L. E., Mahdavi, H., Honavar, V., & Morgan, D. (2024, September 9). Regression with Large Language Models for Materials and Molecular Property Prediction. arXiv.](http://arxiv.org/abs/2409.06080)
>
> [Rubungo, A. N., Arnold, C., Rand, B. P., & Dieng, A. B. (2023, October 21). LLM-Prop: Predicting Physical And Electronic Properties Of Crystalline Solids From Their Text Descriptions. arXiv.](http://arxiv.org/abs/2310.14029)

---

> ### Author Response · Authors · 2024-11-26
> **Rebuttal on "limited contributions over prior work"**
>
> > Limited contributions over prior work. Relationship with MatBench should be clearly characterized very early in the paper, due to the similarity. MatBench is designed to benchmark any type of model on the same task, and the main contributions of this work over MatBench are: additional input representations (since this is the main challenge in using LLMs for this type of modeling) and software to help with evaluation and converting between representation formats. Because of the limited empirical results (as discussed above), this paper mostly provides a new benchmark, which is mostly composed of existing data/representations, but not many insights based on the benchmark, or other findings demonstrating its usefulness.
>
> We do not only use existing representations, but also proposed novel ones.
>
> Indeed, we decided to base MatText on MatBench to avoid further fragmentation of the ecosystem and because we do not aim to propose novel property benchmarks but rather understand how the representation impacts performance. It is important to realize that to conduct a study that aims to compare different representations, one needs to have fixed, standard implementations for representations, which we provide.
>
> As our work shows, deriving text representations systematically is not trivial. It is only possible thanks to the MatText software package (one could also think of it as Matminer focused on text representations).
>
> In our other comments, we enumerate the concrete technical and scientific contributions.

---

> ### Author Response · Authors · 2024-11-26
> **Rebuttal on "limited analysis"**
>
> > Limited analysis of results. The appendix contains the beginning of some interesting analyses of the results. In particular, I would recommend moving discussion of the experimentation with tokenization to the main paper in future drafts, and elaborating. I’m very surprised that changes to the tokenization did not meaningfully improve performance. It would be interesting to see some examples of the tokenized inputs, under the default and modified settings. The encoded representations are so different than the text the default tokenizers were trained on, I would expect this to have a significant impact, and I wonder if just implementing Born & Manica (2023) was insufficient.
>
> We are glad to read that some of our results surprised you. We think that this is a testament to the importance and value of our analysis.
> Importantly, we pretrained the BERT models from scratch with the new tokenizers. Hence "The encoded representations are so different than the text the default tokenizers were trained on" might not be fully precise. We have now included examples for tokenized representations in the revised version of the manuscript.
>
> > Discussion of limitations could be improved. This work has many limitations not included. Breadth of models compared was limited. For example, benefits of scaling are only discussed in the context of “grokking” but it’s not clear why scaling more would not help further, since it was shown to lead to improvements in the existing results. The BERT models are incredibly small (4 layers and 8 attention heads), etc.
>
> Indeed there are limits to the work we conducted. With more than 980 training runs, we decided that we cover many of the most commonly used regimes of LLMs. This number also includes the additional scaling experiments with LLama2-7B, LLama2-13B and LLama2-70B we added in the revised version. Those scaling results underline our previous findings with only marginal benefits of scaling --- likely because the issues are deeper (as highlighted by our analysis with the hypothetical potential as well as the attention analysis).

---

### Author Response · Authors · 2024-11-26
**Response to reviewers**

We thank all reviewers for their feedback. We would like to highlight the key technical and scientific contributions of our work, which we did not make clear enough in our original draft:

**Technical Contributions:**
1. First comprehensive benchmark of text-based materials modeling:
   - 405 Llama and 540 BERT finetune runs
   - 36 backbone training runs
   - Systematic evaluation across different scales (7B to 70B parameters)
   - Novel Python package with consistent implementations

2. Novel analysis methodology:
   - Hypothetical potential experiments that precisely ablate compositional and geometric contributions
   - Attention analysis showing how models process different types of information
   - Systematic evaluation of tokenization strategies

**Scientific Insights:**
1. *More information is not better:*
   - Geometric information is not effectively leveraged by current text-based approaches
   - Additional information (e.g., CIF) sometimes even degrades performance
   - This limitation persists across different:
     - Representations
     - Model architectures
     - Tokenization strategies
     - Scales (both data and model size)

2. *More scale is not better:*
   - Only marginal benefits from scaling, as shown in our comprehensive scaling experiments:

| Task | SLICES | CIF P₁ | Composition | CIF Symmetrized | Z-Matrix | Local-Env | SOTA |
|------|---------|---------|-------------|----------------|-----------|------------|------|
| **BERT Custom Number Tokenizer** ||||||||
| Shear Modulus (GPa) | 0.144±0.002 | 0.152±0.003 | 0.175±0.006 | 0.152±0.007 | 0.153±0.002 | 0.169±0.008 | 0.108±0.001 |
| Bulk Modulus (GPa) | 0.149±0.003 | 0.154±0.007 | 0.173±0.006 | 0.153±0.008 | 0.152±0.002 | 0.154±0.007 | 0.104±0.004 |
| Perovskites (eV) | 0.099±0.007 | 0.095±0.007 | 0.563±0.006 | 0.109±0.008 | 0.095±0.005 | 0.098±0.009 | 0.055±0.004 |
| Bandgap (eV) | 0.843±0.009 | 1.565±0.033 | 0.959±0.010 | 1.125±0.030 | 1.095±0.013 | 0.805±0.005 | 0.396±0.005 |
| Formation Energy (eV) | 0.349±0.017 | 0.681±0.009 | 0.407±0.008 | 0.358±0.020 | 0.826±0.017 | 0.255±0.007 | 0.048±0.06 |
| Is Metal (ROC AUC) | 0.931±0.002 | 0.920±0.005 | 0.929±0.002 | 0.932±0.002 | 0.875±0.004 | 0.934±0.004 | 0.952±0.007 |
| **BERT Regression Transformer Number Tokenizer** ||||||||
| Shear Modulus (GPa) | 0.150±0.006 | 0.136±0.004 | 0.174±0.005 | 0.153±0.002 | 0.154±0.005 | 0.146±0.002 | 0.108±0.001 |
| Bulk Modulus (GPa) | 0.144±0.003 | 0.143±0.004 | 0.173±0.007 | 0.156±0.006 | 0.154±0.006 | 0.143±0.003 | 0.104±0.004 |
| Perovskites (eV) | 0.101±0.005 | 0.098±0.007 | 0.561±0.007 | 0.108±0.010 | 0.097±0.002 | 0.096±0.005 | 0.055±0.004 |
| **Llama-3-8B-Instruct - Finetuned** ||||||||
| Shear Modulus (GPa) | 0.288±0.076 | 0.343±0.130 | 0.365±0.152 | 0.342±0.132 | 0.382±0.038 | 0.329±0.071 | 0.108±0.001 |
| Bulk Modulus (GPa) | 0.460±0.246 | 0.315±0.077 | 0.302±0.108 | 0.402±0.182 | 0.219±0.017 | 0.480±0.004 | 0.104±0.004 |
| Perovskites (eV) | 0.294±0.150 | 0.181±0.018 | 0.716±0.033 | 0.225±0.031 | 0.286±0.045 | 0.410±0.045 | 0.055±0.004 |
| **Llama-2-7b-chat-hf - Finetuned** ||||||||
| Shear Modulus (GPa) | 0.189±0.006 | 0.194±0.007 | 0.212±0.008 | 0.191±0.008 | 0.193±0.007 | 0.181±0.007 | 0.108±0.001 |
| Bulk Modulus (GPa) | 0.181±0.012 | 0.185±0.011 | 0.196±0.006 | 0.186±0.018 | 0.186±0.008 | 0.177±0.005 | 0.104±0.004 |
| Perovskites (eV) | 0.155±0.010 | 0.130±0.004 | 0.691±0.018 | 0.289±0.008 | 0.146±0.006 | 0.139±0.008 | 0.055±0.004 |
| **Llama-2-13b-chat-hf - Finetuned** ||||||||
| Shear Modulus (GPa) | 0.179±0.007 | 0.180±0.005 | 0.210±0.007 | 0.184±0.009 | 0.178±0.007 | 0.183±0.009 | 0.108±0.001 |
| Bulk Modulus (GPa) | 0.168±0.006 | 0.172±0.009 | 0.197±0.013 | 0.181±0.010 | 0.173±0.009 | 0.163±0.012 | 0.104±0.004 |
| Perovskites (eV) | 0.133±0.005 | 0.227±0.035 | 0.689±0.016 | 0.206±0.014 | 0.146±0.006 | 0.125±0.004 | 0.055±0.004 |
| **Llama-2-70b-chat-hf - Finetuned** ||||||||
| Shear Modulus (GPa) | 0.173±0.008 | 0.180±0.010 | 0.205±0.010 | 0.184±0.009 | 0.182±0.006 | 0.180±0.007 | 0.108±0.001 |
| Bulk Modulus (GPa) | 0.173±0.008 | 0.171±0.008 | 0.193±0.008 | 0.182±0.012 | 0.174±0.012 | 0.171±0.012 | 0.104±0.004 |
| Perovskites (eV) | 0.112±0.002 | 0.115±0.002 | 0.715±0.014 | 0.149±0.006 | 0.127±0.001 | 0.110±0.001 | 0.055±0.004 |

3. *Locality matters:*
   - Local-Env representation performs comparably to or better than representations with explicit positional information
   - Suggests current approaches may be most effective when problems can be solved using local contributions

These findings provide crucial guidance for the development of more effective approaches to materials modeling using language models and highlight fundamental limitations that need to be addressed.

---

> ### Author Response · Authors · 2024-11-30
>
> Dear Reviewers,
>
> We understand that you are likely reviewing multiple papers; however, as we near the end of the discussion period, we would greatly appreciate your feedback on our rebuttals. We are eager to address any remaining concerns or questions to further improve our paper.
>
> Thank you very much!

---

> ### Author Response · Authors · 2024-12-02
>
> Dear reviewers,
>
> It's been a few days since we posted our rebuttals, and we sincerely hope you had time to read them. We appreciate that you are busy and likely reviewing multiple other papers; however, as we approach the end of the discussion period, your feedback on our rebuttals is critically important to us. We believe our rebuttals have addressed your main concerns and are happy to tackle any remaining issues or questions to further improve the paper.
>
> Thank you very much!

---

### Meta-Review · Area_Chair_tvoc · 2024-12-15

**Metareview:**

The paper investigates the application of LLMs to materials modeling, focusing on a variety of backbones, scales, and tokenization strategies. The study reveals that current LLMs encounter challenges in effectively utilizing geometric information, instead relying predominantly on local features.

Strengths:
The paper addresses a critical challenge in materials modeling using LLMs, particularly the limitations in leveraging geometric information.

Weaknesses:
The claims regarding encoding information and scaling are not sufficiently supported by the evidence presented. For instance, Reviewer VmTz notes that the discussion on encoding information does not adequately account for prior research. Additionally, Reviewer nB8q observes that the paper does not address mechanisms or architectural changes that could substantiate these claims. Furthermore, Reviewers VmTz and CyJR indicate that the arguments related to scaling lack robust logical or experimental backing. Finally, Reviewer q6NL suggests that the paper's interpretation of its limitations may be overly strong.

Conclusion:
The reviewers have unanimously rated the paper as either "Reject" or "Marginally below the acceptance threshold." Based on this feedback, which underscores the need for stronger evidence and a more comprehensive discussion of the claims, I give the rating of "Reject."

**Additional Comments On Reviewer Discussion:**

The reviewers pointed out that the claims related to encoding information and scaling lack sufficient supporting evidence. Despite the authors' responses to the reviewers' concerns, it remains difficult to consider these issues as having been adequately addressed. For example, while improving model performance through scaling requires various considerations such as model size, training compute, and pretraining data size, as pointed out in Reviewer CyJR's review, the authors make an overly strong claim that "more scale is not better," based solely on the experiments they conducted, which show only marginal differences in performance when scaling. Therefore, the final decision was not influenced by the discussion.

---

### Decision · Program_Chairs · 2025-01-22

Reject